# The Effect of Uni-Hemispheric Dual-Site Anodal tDCS on Brain Metabolic Changes in Stroke Patients: A Randomized Clinical Trial

**DOI:** 10.3390/brainsci13071100

**Published:** 2023-07-20

**Authors:** Somaye Azarnia, Kamran Ezzati, Alia Saberi, Soofia Naghdi, Iraj Abdollahi, Shapour Jaberzadeh

**Affiliations:** 1Department of Physiotherapy, Iranian Research Centre on Aging, University of Social Welfare and Rehabilitation Sciences, Tehran 19857-13834, Iran; azarnia.pt.82@gmail.com; 2Neuroscience Research Centre, Poorsina Hospital, Faculty of Medicine, Guilan University of Medical Sciences, Rasht 41937-13111, Iran; 3Department of Physiotherapy, Faculty of Rehabilitation, Tehran University of Medical Sciences, Tehran 65111-11489, Iran; naghdi@sina.tums.ac.ir; 4Department of Physiotherapy, Faculty of Rehabilitation, University of Social Welfare and Rehabilitation Sciences, Tehran 19857-13834, Iran; 5Department of Physiotherapy, Faculty of Medicine, Nursing and Health Sciences, Monash University, Melbourne, VIC 3800, Australia; shapour.jaberzadeh@monash.edu

**Keywords:** transcranial direct current stimulation, metabolism, stroke, magnetic resonance spectroscopy

## Abstract

Uni-hemispheric concurrent dual-site anodal transcranial direct current stimulation (UHCDS a-tDCS) of the primary motor cortex (M_1_) and the dorsolateral prefrontal cortex (DLPFC) may enhance the efficacy of a-tDCS after stroke. However, the cellular and molecular mechanisms underlying its beneficial effects have not been defined. We aimed to investigate the effect of a-tDCS_M1-DLPFC_ on brain metabolite concentrations (N-acetyl aspartate (NAA), choline (Cho)) in stroke patients using magnetic resonance spectroscopy (MRS). In this double-blind, sham-controlled, randomized clinical trial (RCT), 18 patients with a first chronic stroke in the territory of the middle cerebral artery trunk were recruited. Patients were allocated to one of the following two groups: (1) Experimental 1, who received five consecutive sessions of a-tDCS_M1-DLPFC_ M_1_ (active)-DLPFC (active). (2) Experimental 2, who received five consecutive sessions of a-tDCS_M1-DLPFC_ M1 (active)-DLPFC (sham). MRS assessments were performed before and 24 h after the last intervention. Results showed that after five sessions of a-tDCS_M1-DLPFC,_ there were no significant changes in NAA and Cho levels between groups (Cohen’s d = 1.4, Cohen’s d = 0.93). Thus, dual site a-tDCS_M1-DLPFC_ did not affect brain metabolites compared to single site a-tDCS M_1_.

## 1. Introduction

Stroke is the second leading cause of death worldwide [1]. More than 50% of survivors suffer from chronic disability [2]. Motor impairment is the most common physical complication. However, improving motor function in stroke patients remains a challenge [3]. Recently, neurorehabilitation has progressed towards direct brain stimulation, and studies have suggested that brain modulation may have beneficial effects on motor training [4]. Non-invasive brain stimulation (NIBS) aims to transcranially modulate the excitability of specific brain areas [5]. Transcranial direct current stimulation (tDCS) is a form of NIBS that delivers low-intensity direct current through the scalp and facilitates cell plasticity by acting on the neuronal network [6,7,8]. A recent meta-analysis demonstrated the efficacy of tDCS for motor recovery in stroke patients [9].

Changing the parameters of tDCS to achieve the maximum effect is clinically important. One of the most important parameters is electrode placement. Studies have shown that stimulation of brain areas functionally connected to the primary motor cortex (M_1_) increases corticospinal excitability (CSE) [10]. A related method, called uni-hemispheric concurrent dual-site a-tDCS (UHCDS a-tDCS) stimulates two functionally connected brain regions simultaneously [11]. We chose M_1_ and the dorsolateral prefrontal cortex (DLPFC). The DLPFC is largely responsible for attention, executive function, and working memory [12]. There is evidence of a strong link between executive function and the prefrontal cortex [13]. It is possible that DLPFC stimulation in addition to M_1_ has an additive effect on motor recovery via functional connectivity to M_1_, which is thought to be stronger than M_1_ stimulation alone.

Neuroimaging evidence suggests that changes in neuronal and glial metabolism may play an important role in both functional decline and recovery of brain function. Proton magnetic resonance spectroscopy (H-MRS) can detect changes in the metabolic levels of neurotransmitters such as N-acetyl aspartate (NAA), choline (Cho), and creatine (Cr), and can provide a good picture of the metabolic state of damaged tissue [14].

N-acetyl aspartate (NAA) is used as a non-invasive marker of neurological health. Stroke survivors have shown decreased levels of brain NAA [15], suggesting a loss of neurons.

NAA deficiency is associated with reduced levels of ATP, acetyl CoA and other metabolites involved in energy metabolism [11]. The researchers found that the recovery of NAA levels was only observed in conjunction with the regeneration of ATP [15]. Cr, found in neurons and glial cells, plays an important role in maintaining the high levels of energy required to maintain membrane potentials [11]. Cho and its metabolites can affect functions such as maintaining the structural integrity of cell membranes and transmembrane signaling [12,13].

Hone-Blanchet et al. showed that anodal tDCS to the left DLPFC and cathodal tDCS to the right DLPFC in healthy subjects had rapid excitatory effects during stimulation and increased the amount of NAA in the left DLPFC [16]. Carlson et al. reported decreases in glutamate/glutamine and Cr after cathodal tDCS compared to sham tDCS [17].

The present study aims to extend the previous MRS research with metabolites in stroke patients. The aim of this study is to investigate whether the addition of DLPFC stimulation to M_1_ (UHCDS a-tDCS_M1-DLPFC_) can alter brain metabolite concentrations. We hypothesized that the levels of brain metabolites such as NAA, creatine, and choline would change significantly after UHCDS a-tDCS_M1-DLPFC_ treatment compared to baseline levels.

## 2. Materials and Methods 

### 2.1. Participants and Study Design

Eighteen patients with a first chronic stroke (>6 months post-stroke) in the MCA territory were enrolled in this double-blind, randomized clinical trial. The study sample was recruited from 533 patients who were admitted to Pars Hospital with a diagnosis of stroke between 20 June 2021 and 20 July 2022, diagnosed by a physiotherapist and a neurologist based on the admission criteria.

Ischemic stroke was confirmed clinically and by neuroimaging. Patients had no history of chronic neurological or cardiac disease and were not taking any medication that could alter their cognitive state. The severity of wrist flexor Spasticity was 1 or higher on the Modified Modified Ashworth Scale (MMAS). They were able to communicate verbally with the therapist. They did not have severe cognitive and memory impairment according to the Persian version of the Mini-Mental State Examination (MMSE) (MMAS ≥ 23). Figure 1 shows the study procedure.

Patients were assured that they could withdraw from the study at any time. All patients gave written informed consent to participate in the study. The study was approved by the Ethics Committee of the University of Social Welfare and Rehabilitation (IR: USWR.REC.1400.185).

#### Randomization

The assessor and the participants were kept blinded to the group allocation. Randomization was carried out using the Randomization.com website (accessed on 20 March 2023). The patients were randomized into two groups: Experimental 1 and Experimental 2, using a computer-generated randomization block. (1) Experimental 1 received five consecutive sessions of a-tDCS M_1_-DLPFC M_1_ (active)-DLPFC (active). (2) Experimental 2 received five consecutive sessions of a-tDCS M_1_-DLPFC M_1_ (active)-DLPFC (sham). All patients were assessed by MRS before and 24 h after five consecutive sessions of tDCS intervention. All patients completed the intervention period and there were no dropouts. Figure 1 demonstrates the CONSORT flow diagram depicting the phases of enrollment, intervention allocation, follow-up, and data analysis in this two-group parallel randomized trial (Figure 1).

### 2.2. H-MRS Protocol

MRS data were acquired using a Siemens 1.5 T scanner (Erlangen, Germany) with an eight-channel receive-only head coil. A conventional 3-dimensional brain image (sagittal T1 MPRAGE, TR/TE = 1800/3.5, field of view (FOV) = 256 × 256 × 160 mm^3^, resolution = 1 × 1 × 1 mm^3^) was acquired for all patients before the MRS sequence as a reference image for volume of interest (VOI) positioning. For single-voxel spectroscopy (SVS), MRS was acquired using a point-resolved spectroscopy (PRESS) sequence. Two 2 × 2 × 2 cm^3^ voxels were located in the primary motor cortex (M_1_), dorsolateral prefrontal cortex (DLPFC). Voxels were carefully placed to avoid contact with subcutaneous fat, skull, vasculature, arachnoid space, and cerebrospinal fluid. Manual shimming was performed on all acquisitions. Parameters were set to TR/TE = 1500/135 and NEX = 128. Six saturation bands were placed around the VOI to suppress external volume signals. The average duration of each H-MRS acquisition was 10 ± 2 min (5 min for each region) with no complications.

#### MRS Data Processing

Data were pre-processed by applying a water removal algorithm to the reference offset of 4.65 ppm to remove residual water signals. SVS raw data were fitted using TARQUIN (Gerg Reynolds and Martin Wilson, version 4.3.10). The predefined data set of NAA, Cho, and Cr target metabolites was selected for peak fitting and metabolite concentration. The metabolite ratios of NAA/Cr and Cho/Cr were calculated by dividing the metabolite values in the same spectrum for the M_1_ region.

### 2.3. Transcranial Direct Current Stimulation

Two single-channel tDCS devices delivered direct current stimulation through two saline-soaked electrodes. Electrode placement was determined using the international 10–20 system of electroencephalography. In both groups, the active electrodes were placed on M_1_ (C3/C4) and DLPFC (F3/F4) according to the involved hemisphere, and the reference electrodes were placed on the supraorbital area of the uninvolved side (Figure 2) [10]. According to the previous research [18], a constant current of 1 mA was applied for 20 min. In the sham group “experimental 2”, the stimulation was switched off after 30 s only in the DLPFC region. The standard 5 × 7 cm^2^ electrode was used as the reference electrode. To localize the excitability of the motor cortex and increase the excitability of the corticospinal tract, an active electrode of 4 × 4 cm^2^ was applied to the M_1_ and DLPFC regions [10,19].

Ref. [20] Schematic illustration of electrode montage in experimental 1: UHCDS a-tDCS_M1-DLPFC_ and experimental 2: UHCDS a-tDCS_M1-DLPFC_ (M_1active_-DLPFC _sham_); The reference electrodes were placed over the contralateral supraorbital area in two conditions. In both groups, the active electrodes were positioned over M_1_ and dorsolateral prefrontal cortex (DLPFC).

### 2.4. Measurement of Metabolites

MRS is an objective, non-invasive technique to detect and quantify changes in certain biochemical compounds such as NAA, Cr, and Cho in brain tissue. MRS data were collected from M_1_ for all patients.

### 2.5. Experimental Procedures

The study procedures consisted of three steps: baseline assessment, intervention period, and post-intervention period. In the first step, MRS data were collected from patients in both groups at baseline. In the next step, all patients received five sessions of tDCS according to the group allocation.

The stimulation dose was selected based on a previously published study. In the [10,18] post-intervention period, patients underwent MRS 24 h after the last tDCS session (Figure 3).

### 2.6. Outcome Measures and Data Analysis

The primary outcome was the concentration of brain metabolites (NAA, Cr, Cho) and the metabolite ratio (NAA/Cr, Cho/Cr) in M_1_ tested by H-MRS. Metabolite levels on local brain H-MRS are often reported as ratios rather than absolute concentrations. The most common denominator is the Cr level, which is thought to be stable under normal conditions as well as under some pathological conditions [21]. Therefore, we examined NAA/Cr and Cho/Cr.

Data analysis was performed using SPSS software version 26 (IBM SPSS Statistics for Windows, version 26, IBM Corp, Armonk, NY, USA). Continuous variables were summarized as mean ± standard deviation. The Shapiro-Wilk test was used to determine the normal distribution of quantitative data. The test results indicated that the MRS data were not normally distributed. Non-parametric Mann-Whitney U test and Wilcoxon signed rank test were used to compare MRS data between/within groups.

Group differences were examined by ANCOVA controlling for baseline metabolite. *p* < 0.05 was considered statistically significant. The sample size was calculated using G*Power software (version 3:1, Heinrich-Heine-University) based on the effect size (d = 2.0) derived from the Rayen study (power of 0.90 and α = 0.05). We compensated for 20% of the dropouts.

## 3. Results

Eighteen stroke patients (10 female, 8 male) with a mean age of 60.94 ± 6.92 years were enrolled. The mean time since stroke onset was 34.28 ± 8.91 weeks. Table 1 shows that there were no statistical differences between the two study groups in terms of demographic characteristics, comorbidities, and spasticity level. This study assessed the mean NAA, Cr, Cho, NAA/Cr, and NAA/Cho between/within the two groups at baseline and after intervention in M_1_.

### 3.1. Between-Group Comparison

The results showed significantly higher NAA and Cho concentrations in M_1_ after the intervention (*p* = 0.040, *p* = 0.050 respectively), with large effect sizes for NAA and Cho, 1.41 and 0.93 respectively. Metabolite ratio results showed a non-significant difference in NAA/Cr and Cho/Cr after intervention (*p* = 0.113, *p* = 0.387).

### 3.2. Comparison within Groups

The result showed significant changes in NAA, Cr, and Cho in group Experimental 2 (*p* = 0.008), and the concentration of metabolites was increased. In group Experimental 1 there were significant differences in Cr. Cr concentration was decreased (Table 2). For changes in metabolite ratios, there was a significant difference in NAA/Cr in both groups. However, changes in Cho/Cr (*p* = 0.004) were only observed in group Experimental 2 (Figure 4).

## 4. Discussion

To our knowledge, this is the first study investigating changes in brain metabolites after uni-hemispheric concurrent dual-site a-tDCS in chronic stroke patients.

The main findings of the results were significantly higher NAA, Cr, and Cho concentration in the M_1_, in the group single-site a-tDCS_M1_ compared to a-tDCS_M1-DLPFC_, as measured by 1.5 T MR spectroscopy.

Previous literature has investigated bi-hemispheric single-site tDCS in healthy subjects [16,21,22] and children with spastic cerebral palsy (CP) [23,24]. Studies have shown that a-tDCS increases the levels of NAA and Cho [16,24]. Our study was also consistent with the previous study, and the group Experimental 2 that received the single-site stimulation had a significant increase in metabolites after the intervention. Hone-Blanchet et al. [16] showed that the online effect of a single session of a-tDCS on the DLPFC increased the amount of NAA. Auvichayapat et al. [24] reported an increase in Cr, Cho, and NAA after tDCS in the basal ganglia of CP patients. N-acetyl aspartate is usually considered a neuronal marker because it is only found in mature neurons.

Researchers have found an association between low levels of brain NAA concentration and poor motor function in patients after stroke, and increased levels of NAA were also predictive of recovery [25]. Glodzik-Sobanska et al. showed that an increase in NAA in stroke patients was associated with neurological improvement [24]. Perhaps an increase in NAA after a-tDCS is due to an increase in neuronal excitability leading to long-term potentiation, such as plasticity. However, the study of metabolites in dual-site stimulation has not been investigated. Previous fMRI studies have shown that dual-site stimulation increases corticospinal excitability up to twofold [26,27].

Our results also showed an increase in Cho concentration in both groups, particularly significant in Experimental group 2. This finding is consistent with Auvichayapat et al. [24] Cho is a membrane marker and its metabolites play an important role in a variety of mechanisms, such as maintaining the structural integrity of the cell membrane, methyl metabolism, and transmembrane signaling. In this case, choline repletion may affect neuronal connections and facilitate neuroplasticity in the adult CNS.

The lesser increase of NAA and Cho in the group receiving dual-site a-tDCS of both the DLPFC and M_1_ region could be explained by the concept of homeostasis—that is, the ability of the human brain to regulate changes in synaptic plasticity to avoid drastic changes in its function. Homeostasis maintains stable function against changes in the activity of the number and strength of synapses. Homeostatic plasticity is increasingly recognized as a regulator of neural change within physiological limits [24]. In this context, researchers emphasize homeostatic plasticity as a tool to prevent the instability of the neural network that occurs in neurorehabilitation. Thus, dual-site stimulation could not induce further changes by overshooting the physiological range.

## 5. Limitations

The limitations of this study should also be noted. Firstly, changes in brain metabolites were measured only 24 h after the last stimulation session, and at longer follow-up times or immediately after the intervention. Therefore, we were not able to investigate immediate and long-term effects. Secondly, a single voxel MRS was used with a 1.5 T MRI, which may have limited the collection of data from multiple brain regions simultaneously. It could be suggested that further studies use a multi-voxel 3 or 7 T MRI system to measure stimulation effects in multiple brain regions, and to investigate other metabolites. Thirdly, the tDCS intervention consisted of five consecutive days of 20 min tDCS applications, may not be sufficient to alter brain metabolites. Finally, chronic stroke patients were included in the current study, so it is suggested that future studies investigate the changes in metabolites in subacute patients and examine the levels of metabolites in both hemispheres.

## 6. Conclusions

This study aimed to investigate the effect of adding transcranial direct stimulation of the DLPFC to M_1_ stimulation on changes in brain metabolites in the M_1_ region. The results showed that there are no significant changes in the amount of brain metabolites after UHCDS a-tDCS_M1-DLPFC_ compared to a-tDCS M_1_.

## Figures and Tables

**Figure 1 brainsci-13-01100-f001:**
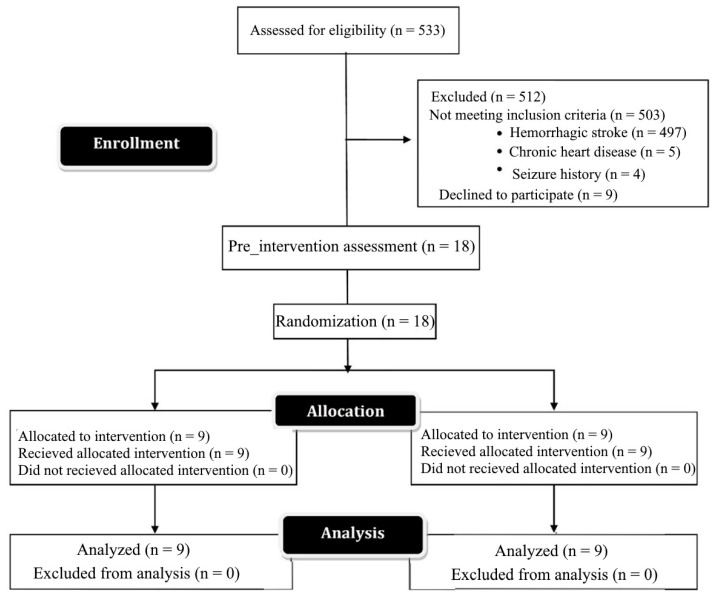
Consort diagram of patients.

**Figure 2 brainsci-13-01100-f002:**
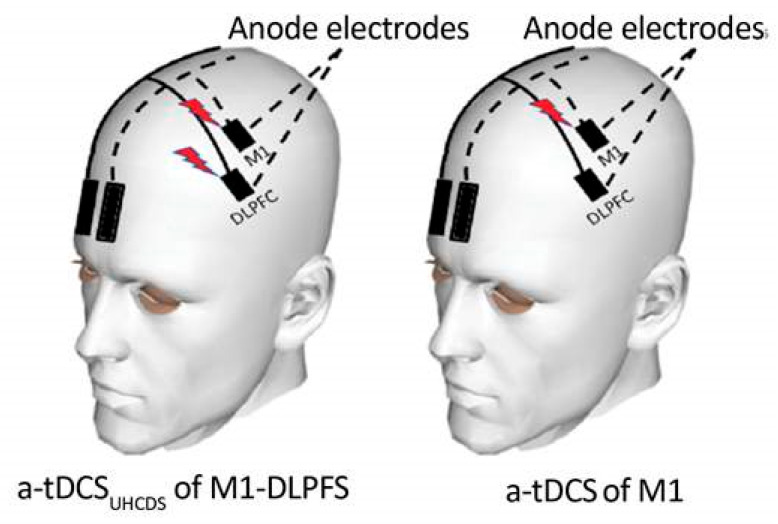
This figure is adapted from (The effects of anodal-tDCS on corticospinal excitability enhancement and its after-effects: Conventional vs. uni-hemispheric concurrent dual-site stimulation, Vaseghi et al., 2015 [10]).

**Figure 3 brainsci-13-01100-f003:**
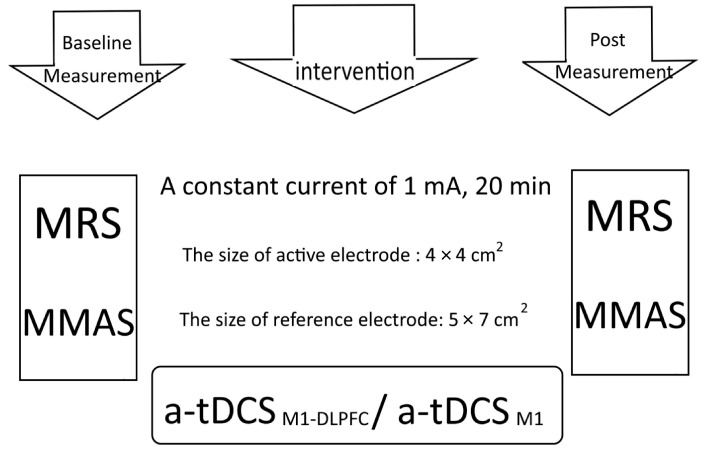
Experimental design for the comparison of conventional a-tDCS and UHCDS a-tDCS M_1_-DLPFC. MMAS: Modified Modified Ashworth Scale, MRS: Magnetic resonance spectroscopy.

**Figure 4 brainsci-13-01100-f004:**
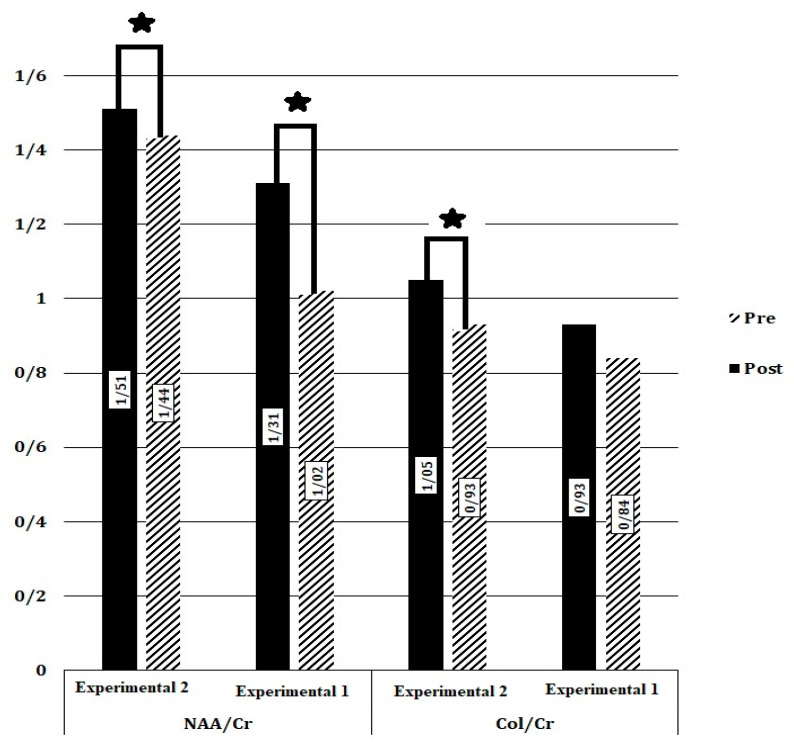
Metabolite ratio (NAA/Cr, Col/Cr) changes in M_1_. Data are presented as the mean of brain metabolites compared between baseline and post-tDCS intervention. ★: Significant difference.

**Table 1 brainsci-13-01100-t001:** Summary of the Clinical Data.

		Study Group			
		Experimental 1 (n = 9)	Experimental 2 (n = 9)	Total (n = 18)	*p*-Value
		N(%)	N(%)	N(%)	
Gender	Female	3(33.3)	7(77.8)	10(55.6)	0.15 *
	Male	6(66.7)	2(22.2)	8(44.4)	
Age	Mean(SD)	61.8(6.25)	60.0(7.79)	60.9(6.92)	0.57 †
Weeks since stroke	Mean(SD)	32.4(4.88)	36.1(11.72)	34.2(8.91)	0.39 †
MMSE	Med (range)	30(29,30)	30(29,30)	30(29,30)	0.99 ‡
MMAS	Med (range)	2(1,4)	1(1,2)	2(1,4)	0.06 ‡
Comorbidity diseases					
Hypertension	Yes	4(44.4)	3(33.3)	7(38.9)	0.99 *
Diabetes mellitus	Yes	2(22.2)	4(44.4)	6(33.3)	0.62 *
Dyslipidemia	Yes	3(33.3)	2(22.2)	5(27.8)	0.99 *

* Fisher’s Exact Test; † Independent T-Test; ‡ Mann-Whitney U.

**Table 2 brainsci-13-01100-t002:** Differences between baseline and post-intervention of brain metabolites in the M_1_ between two groups.

Brain Metabolites	Time	Mean(SD)		*p*-Value	
		Experimental 1	Experimental 2	*p*-value †	*p*-value *
**NAA**	Baseline	158.4(59.6)	194.1(12.4)	0.43	0.04
	Post-intervention	190.9(35.3)	229.2(14.5)	0.04	
	*p*-Value	0.08	0.008		
**Cr**	Baseline	166.73(25.21)	135.23(14.60)	0.004	0.04
	Post-intervention	146.95(12.15)	152.58(15.15)	0.43	
	*p*-Value	0.02	0.008		
**Cho**	Baseline	142.98(23.58)	126.09(23.98)	0.22	0.008
	Post-intervention	137.62(20.07)	159.39(26.27)	0.05	
	*p*-Value	0.59	0.008		

* ANCOVA; † Mann-Whitney U.

## Data Availability

Data available on request from the authors.

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
