# Peer review of "The Effect of Uni-Hemispheric Dual-Site Anodal tDCS on Brain Metabolic Changes in Stroke Patients: A Randomized Clinical Trial"

_brainsci, 2023, doi:10.3390/brainsci13071100_

Round 1

Reviewer 1 Report

Comments and Suggestions for Authors

The title is well presenting the main focus of the present manuscript, as the abstract is well structured, informative, describing well the whole manuscript and the final conclusions, but I would recommend to remove that the study was by the University of Social Wel- 34 fare and Rehabilitation Ethics Committee - this is not so important for the abstract - let it be in the material and methods.

The introduction makes a good review (background) on the topic, as the information is relevant to the study. Nevertheless, it seems a little bit too long, so it can be optimized in order to hold the interest of the reader.

The authors tried to describe the material and methods in relatively good manner, but it can definitely be improved by: removing Table 1 - it presents results, even if it is related with the material - add it in the results section. Also add information about the MRS scale that you use in this section. 

The presented results are interesting, they follow the line of some previous studies, but additionally they give light at the focused issue. Nevertheless, there are some minor  things to improve such as "cm2" should be fixed with superscript of "2" and other minor technical issues, so please double check it again.  Again you should try to make better differentiation between the material and methods section and the results. Otherwise, all the information presented is described well in a interesting manner with good illustrations. 

At the discussion the authors present up-to-date information which is relevant with the previous literature. No further recommendations.

Based on the presented results and relations with the current literature - the conclusions are relevant and in correlation with the general workflow of the manuscript. Also the authors presented relevant limitations of their study, and of course further follow up and larger clinical group may put more light on this issue.

Regarding the references, there are newer articles about some of the mentioned issues, for example: Feigin, V.L.; Lawes, C.M.; Bennett, D.A.; Barker-Collo, S.L.; Parag, V. Worldwide stroke incidence and early case fatality re- 271 ported in 56 population-based studies: a systematic review. The Lancet Neurology 2009, 8, 355-369 -> there is newer publication again by prof. Feigin so it can be updated. Double check again the other citations.

Comments on the Quality of English Language

The manuscript requires double check for English grammar and punctuation.

Author Response

Dear professors and judges

Greetings and respect, while thanking and appreciating the great attention and accuracy that you have had in reading this article and identifying the existing flaws and defects, It is noteworthy that the most important issues considered by the esteemed judges in the article are stated in the table below. In addition, the desired corrections have been applied in red in the text of the article.

We hope that these revisions have improved the paper such that you now deem it worthy of publication.

Comments of the Reviewer 1

Authors' answers

The title is well presenting the main focus of the present manuscript, as the abstract is well structured, informative, describing well the whole manuscript and the final conclusions, but I would recommend to remove that the study was by the University of Social Welfare and Rehabilitation Ethics Committee - this is not so important for the abstract - let it be in the material and methods.

Thank you very much for your kind comment - it has been corrected.

The introduction makes a good review (background) on the topic, as the information is relevant to the study. Nevertheless, it seems a little bit too long, so it can be optimized in order to hold the interest of the reader.

Out of respect for your opinion, there have been a few changes in line with the journal's word count.

The authors tried to describe the material and methods in relatively good manner, but it can definitely be improved by: removing Table 1 - it presents results, even if it is related with the material - add it in the results section. Also add information about the MRS scale that you use in this section. 

The proposed amendment has been made. Section 2-2 provides information on MRS

The presented results are interesting, they follow the line of some previous studies, but additionally they give light at the focused issue. Nevertheless, there are some minor things to improve such as "cm2" should be fixed with superscript of "2" and other minor technical issues, so please double check it again. Again you should try to make better differentiation between the material and methods section and the results. Otherwise, all the information presented is described well in a interesting manner with good illustrations. 

The changes made are highlighted. Technical issues have been reviewed

Reviewer 2 Report

Comments and Suggestions for Authors

An interesting study using a novel tDCS protocol in stroke patients with MRS to analyse alterations in metabolites.  In general the manuscript is well written.

Comments and Suggestions:

The end of the abstract should not include a statement about ethics but rather be used to interpret the results and provide a short discussion of the wider implications. This should be modified. I don't believe the ethical statement is necessary in the abstract - it can be in the methods - but if it is included then it should be early in the experiment when outlining the participant group.

The introduction and discussion both could be enhanced to improve the relevancy of the manuscript. For instance, specifically to include studies that suggest NAA as biomarker of stroke recovery. Also to discuss the relevance of alterations in these metabolites to functional improvements. What is the evidence?

L45: why are quotes used here ?

Has any modelling work of current flow been performed on the montage used in this study?

Figure 1: Please clarify 'consort' diagram. I am not familiar with this term.

Figure 2: Include the units for the size of the reference electrode.

The methods section could include justification for specific electrode sizes in this study.  In this montage the electrodes are quite close together so it is important to discuss (again modelling data would be useful here).

The choice of statistical tests is not entirely clear - since the authors report the data is non parametric but do include ANCOVA.

The sample size is quite small and MRS is not measured in a matched group without any tDCS

L204: capitalization missing at the start of the sentence

L208: the authors do not include a citation but should - 'studies have shown tDCS increases...' etc  What studies? Provide at least two.

L217: the authors state previous fMRI studies but do not cite the reference here - it should be included.

L219: the authors should only state significant results

Author Response

Dear professors and judges

Greetings and respect, while thanking and appreciating the great attention and accuracy that you have had in reading this article and identifying the existing flaws and defects, It is noteworthy that the most important issues considered by the esteemed judges in the article are stated in the table below. In addition, the desired corrections have been applied in red in the text of the article.

We hope that these revisions have improved the paper such that you now deem it worthy of publication.

Comments of the Reviewer 1

Authors' answers

An interesting study using a novel tDCS protocol in stroke patients with MRS to analyse alterations in metabolites.  In general the manuscript is well written.

Thank you very much for your kind comment.

The end of the abstract should not include a statement about ethics but rather be used to interpret the results and provide a short discussion of the wider implications. This should be modified. I don't believe the ethical statement is necessary in the abstract - it can be in the methods - but if it is included then it should be early in the experiment when outlining the participant group.

The appropriate changes have been performed.

line 33, line 99

The introduction and discussion both could be enhanced to improve the relevancy of the manuscript. For instance, specifically to include studies that suggest NAA as biomarker of stroke recovery. Also to discuss the relevance of alterations in these metabolites to functional improvements. What is the evidence?

it has been Corrected and highlighted in colour.

Page 2, line 64-68

Page9, line 240-242, 252,

 L45: why are quotes used here ?

L204: capitalization missing at the start of the sentence

L208: the authors do not include a citation but should - 'studies have shown tDCS increases...' etc What studies? Provide at least two.

L217: the authors state previous fMRI studies but do not cite the reference here - it should be included.

It has been Corrected and highlighted in colour

Has any modelling work of current flow been performed on the montage used in this study?

The methods section could include justification for specific electrode sizes in this study.  In this montage the electrodes are quite close together so it is important to discuss (again modelling data would be useful here).

We did not model in our study but the article " J. S. A. Lee· S. Bestmann · C. Evans. A Future of Current Flow Modelling for Transcranial Electrical Stimulation? " has investigated the modeling of transcranial stimulation flow.

According to the article " Paula Faria, Mark Hallett, Pedro Cavaleiro Miranda. A finite element analysis of the effect of electrode area and interelectrode distance on the spatial distribution of the current density in tDCS ", smaller electrodes produce a more focal current density distribution in the brain.

The small size of active electrode produces a highly focused DC current over the target areas, which enabled us to stimulate M1 and S1 with two separated anode electrodes separately. Based on the result of some computational modeling studies, the effects of tDCS can be more focalized by smaller electrodes (Nitsche et al., 2007; Bikson et al., 2010). In addition, recent experimental investigations on human brain illustrated that utilizing smaller active electrodes over M1 resulted in larger CSE (Nitsche et al., 2007; Bastani and Jaberzadeh, 2013; Vaseghi et al., 2015b).

Page 5, line 148-150

Figure 1: Please clarify 'consort' diagram. I am not familiar with this term.

CONSORT flow diagram of the progress through the phases of a parallel randomized trial of two groups (that is, enrolment, intervention allocation, follow-up, and data analysis)

Figure 2: Include the units for the size of the reference electrode.

It has been Corrected

The choice of statistical tests is not entirely clear - since the authors report the data is non parametric but do include ANCOVA.

According to the article “Olejnik SF, Algina J. Parametric ANCOVA and the rank transform ANCOVA when the data are conditionally non-normal and heteroscedastic. Journal of Educational Statistics. 1984 Jun;9(2):129-49”:

The results showed that parametric ANCOVA was robust to violations of either normality or homoscedasticity.

Because according to Lone's test.

Levene's test for equality of error variance

Our data are homoscedastic, so we can use parametric ANCOVA.

The sample size is quite small and MRS is not measured in a matched group without any tDCS

We calculated the sample size based on the Rayen study (effect size (d=2.0), power of 0.90 and α=0.05). page 6, line 192.

According to "Bridges Nair: Clinical Trial Designs" article, One of the methods of selecting the control group is active treatment concurrent control, where a new treatment method is compared with the standard method.

The Declaration of Helsinki requires the use of standard treatment as a control.

According to the article "Felipe Fregni; Mirret M. El-Hagrassy; Kevin Pacheco-Barrios; Sandra Carvalho;

Jorge Leite. Evidence-Based Guidelines and Secondary Meta-Analysis for the Use of Transcranial Direct Current Stimulation in Neurological and Psychiatric Disorders", a-tDCS has a level B recommendation for motor rehabilitation in chronic stroke patients, so according to the guidelines, we considered the a-tDCS group single site M1 region as the

control group.